# l-Cysteine and Vitamin D Co-Supplementation Alleviates Markers of Musculoskeletal Disorders in Vitamin D-Deficient High-Fat Diet-Fed Mice

**DOI:** 10.3390/nu12113406

**Published:** 2020-11-06

**Authors:** Rajesh Parsanathan, Arunkumar E. Achari, Prasenjit Manna, Sushil K. Jain

**Affiliations:** Department of Pediatrics and Center for Cardiovascular Diseases and Sciences, Louisiana State University Health Sciences Center, 1501 Kings Highway, Shreveport, LA 71130, USA or rajesh.uom@gmail.com (R.P.); bioarun1985@gmail.com (A.E.A.); pmanna2012@gmail.com (P.M.)

**Keywords:** vitamin D deficiency, l-cysteine, glutathione, myogenic markers, dystrophy markers, skeletal muscle

## Abstract

Vitamin D (VD) deficiency is associated with musculoskeletal disorders. This study examines whether co-supplementation of l-cysteine (LC) and VD is better than monotherapy with LC or VD at alleviating musculoskeletal dyshomeostasis in the skeletal muscle of VD-deficient high-fat diet (HFD-VD-) fed mice. Mice were fed a healthy diet or an HFD; for VD-deficient animals, the mice were maintained on a HFD-VD-diet (16 weeks); after the first 8 weeks, the HFD-VD-diet-fed mice were supplemented for another 8 weeks with LC, VD-alone, or the same doses of LC + VD by oral gavage. Saline and olive oil served as controls. Myotubes were exposed with high-glucose, palmitate, Monocyte Chemoattractant Protein 1 (MCP-1), and Tumor Necrosis Factor (TNF), to mimic the in vivo microenvironment. In vitro deficiencies of glutathione and hydrogen sulfide were induced by knockdown of GCLC and CSE genes. Relative gene expression of biomarkers (myogenic: MyoD, Mef2c, Csrp3; muscle dystrophy: Atrogin1, Murf1, and Myostatin; bone modeling and remodeling: RANK, RANKL, OPG) were analyzed using qRT-PCR. Co-supplementatoin with LC + VD showed beneficial effects on gene expression of myogenic markers and OPG but reduced markers of dystrophy, RANK/RANKL in comparison to LC or VD alone-supplementation. In vitro myotubes treated with glutathione (GSH) precursors also showed a positive effect on OPG and the myogenesis genes, and inhibited RANK/RANKL and muscle-dystrophy markers. This study reveals that the co-supplementation of LC with VD significantly alleviates the markers of musculoskeletal disorders in the skeletal muscle better than monotherapy with LC or VD in HFD-VD-fed mice.

## 1. Introduction

Vitamin D (VD) deficiency or insufficiency is associated with diseases affecting muscle and bone health [1,2,3]. Low blood levels of both 25-hydroxyvitamin D (25(OH)D) and glutathione (GSH) are positively associated with metabolic syndrome in human subjects [4,5,6,7,8]. Antioxidant GSH deficiency increases the oxidative stress that may favor endogenous protein oxidative modification, impairs cellular physiology, and leads to the disease’s manifestation. Supplementation with GSH or its precursor, the sulfur-containing amino acid l-cysteine (LC), has been successfully used to improve the GSH status in blood and tissues, reducing immune-metabolic syndrome [4,5,6,8]. 

Skeletal muscle is the largest tissue in the body, and any loss of function or regenerative properties debilitates the musculoskeletal system [9]. Myogenic markers such as myoblast determination protein 1 (MyoD), myocyte enhancer factor 2C (Mef2c), and cysteine and glycine-rich protein 3 (Csrp3) are positive regulators and promote myogenesis, regeneration, and play an essential role in muscle function [10,11]. Conversely, skeletal muscle-specific F-box protein (Atrogin1), muscle RING-finger protein-1 (Murf1), and Myostatin (Mstn) are critical molecules involved in muscle atrophy [12,13]. Skeletal muscle dystrophy/atrophy is a debilitating consequence of many pathological conditions and diseases [14]. Receptor activator of nuclear factor-kB (RANK), its ligand RANKL, and the soluble decoy receptor osteoprotegerin (OPG) pathway control bone remodeling and homeostasis [15,16]. The effects of RANK/RANKL/OPG extend well beyond its classical functions; in skeletal muscle, interaction with RANKL/RANK causes atrophy and dysfunction, whereas OPG provides significant protection against muscle damage [15].

Studies have shown that high-fat diet (HFD)-induced obesity leads to skeletal muscle oxidative stress, inflammation, and muscle mass loss by decreasing myogenic markers such as MyoD, Mef2c, and Csrp3, increasing muscle dystrophy markers such as Atrogin1, Murf1, and Myostatin [17,18,19]. Supplementation with a cysteine/thiol-based antioxidant delays or attenuates muscular dysfunction [20,21,22]. Similarly, supplementation with N-acetyl cysteine decreases osteoclast differentiation and increases bone mass in obese diabetic mice [23]. VD also maintains a normal bone resorption rate and formation through the RANKL/OPG signal [24]. VD deficiency is detrimental to muscle function, independent of alterations in phosphate and calcium levels [25]. Observational studies of VD-deficiency also associate reduced muscle mass and weakness [1,2,3]. However, interventional trials and meta-analyses of VD deficiency have yielded contradictory findings [25]. Our previous preclinical studies demonstrate that GSH epigenetically regulates VD metabolism genes. Supplementation with the VD + LC combination was more successful at boosting 25(OH)D levels by improving the status of VD metabolism genes in the liver, kidney, and muscle [4,5,6,7,8,26]. However, as far as we know, no previous study has examined the effect of co-supplementation with LC+VD on musculoskeletal markers in the muscle of HFD-VD-mice. 

This study reports that LC (a GSH precursor) co-supplementation with VD significantly alleviates dyshomeostasis of the skeletal muscle in VD-deficient high-fat diet-fed mice, suggesting that combined supplementation with the nutraceuticals LC + VD could be a better option for musculoskeletal system disorders rather than supraphysiological monotherapy with VD alone.

## 2. Materials and Methods

The reagents used in the study, and all other chemicals were of analytical and molecular grade unless otherwise mentioned, and were purchased from Sigma Chemical Co. (St. Louis, MO, USA).

### 2.1. Animal Experimental Design and Treatment 

Male C57BL/6J mice (5 weeks old, 20–24 g) were procured from The Jackson Laboratory (Bar Harbor, ME, USA). Mice were given access to food and water ad libitum and housed for acclimation (1 week) in a temperature-controlled room (22 ± 2 °C) with light/dark cycles (12/12 h). They were maintained under standard housing conditions throughout the experiment. After receiving approval (P-15-006) from the Institutional Animal Committee, according to the guidelines of the institution’s ethical standards, all the procedures were performed. 

Mice were randomized, labeled in individual cages, and divided into various groups. The mice were fasted overnight and tested for hyperglycemia by measuring their blood glucose concentrations before starting the treatment plan. Fasting blood glucose was analyzed by the tail prick method using a glucometer (Accu-Chek, Boehringer Mannheim Corp., Indianapolis, IN, USA). 

Control animals were fed a healthy diet (Ctrl; lower in fat), while animals in the high-fat diet group were fed a high-fat diet (HFD) for a total of 16 weeks. The mice were maintained on a VD-deficient HFD (HFD-VD-) for 16 weeks (to mimic the VD-deficient condition). After the first 8 weeks, the mice were supplemented by oral gavage for another 8 weeks with either 5 mg LC/kg BW daily (LC), 67 IU VD/kg BW (VD) alone, or the same doses of LC + VD co-supplemented to HFD-VD-mice. Additionally, two control groups of HFD-VD-mice were maintained on oral gavage with either saline (S-Ctrl) or the same dose of the vehicle olive oil used for dissolving cholecalciferol (OO-Ctrl). These diets—(healthy diet, HFD, and HFD-VD-) composition details and dose justification for LC and vitamin D—are given in our recent publication [4]. 

Sex steroids, estrous cycle, and hormonal impacts (sexual dimorphism) influence musculoskeletal markers [27,28,29,30,31]; therefore, we chose only male mice for this pilot study. Furthermore, choosing a 22 ± 2 °C housing temperature better represents humans living in colder regions that also lack environmental temperature controls (to mimic human thermal relation) [32,33].

As mentioned above, at the end of the supplementation, the animals were isoflurane euthanized and then perfused with cold saline. Skeletal muscle (gastrocnemius) was collected immediately, quickly diced, and frozen in liquid nitrogen at −80 °C. This model (HFD and HFD-VD-) of dietary-induced insulin resistance created fasting hyperglycemia, hyperinsulinemia, elevated proinflammatory cytokines, decreased glutathione, and VD deficiency, thus representing a reasonable model of the human condition [4,5,6,34]. 

### 2.2. Cell Culture, Treatments, and RNA Interference of GCLC and CSE

Mouse C_2_C_12_ myoblasts (ATCC^®^ CRL-1772™, Manassas, VA, USA) were cultured and brought to myotubes differentiation with appropriate conditions following the methods used in our previous published studies [5,6,35,36]. High glucose (HG; 25 mM), palmitate (0.6 mM) for 24 h; MCP-1 (2.5 ng/mL) and TNF (250 pg/mL) for 6 h treated to the myotubes, the doses and time points justifications given in our previous published studies [5,6,35,36]. GCLC or CSE and Control siRNA-A (50 and 100 nM siRNA), transiently transfected as per the method described earlier [4,5,6,35,36]. GSH precursor (l-cysteine; 300 µM) or an H_2_S donor (NaHS; 20 µM) were treated following the methods used in our previous published studies [5,6,35,36] to boost the level of GSH cellular content and H_2_S production. Cell viability was determined using the Alamar Blue reduction bioassay in all the experimental conditions [5,6,35,36].

### 2.3. Relative Gene Expression

RNA isolated from cells or tissues, quality and concentration determined, and 1 μg of RNA samples were reverse transcribed, and the qPCR performed as per our previously published studies [5,6,35,36]. The following primer/probe sets from Applied Biosystems™ TaqMan™ Gene Expression Assays were used for qPCR; MyoD (Mm00440387_m1), Mef2c (Mm01340842_m1), Csrp3 (Mm00443379_m1), Atrogin1/Fbxo32 (Mm00499523_m1), MuRF1/Trim63 (Mm01185221_m1), Mstn/Myostatin (Mm01254559_m1), Rank/Tnfrsf11a (Mm00437132_m1), RankL/Tnfsf11 (Mm00441906_m1), Opg/Tnfrsf11b (Mm01205928_m1) and Gapdh (Mm99999915_g1). The results were expressed as the relative quantification (RQ, the fold of control).

### 2.4. Statistical Analyses

Data were generated from multiple repeats of different biological experiments to obtain the mean values and standard errors of the mean. Significance was set at *p* < 0.05, and statistical differences were evaluated using two-way ANOVA followed by Dunnett’s multiple comparisons test between groups were conducted using GraphPad Prism 8.2.1 (GraphPad Software, La Jolla, CA, USA).

## 3. Results

### 3.1. Effect of HFD, VD-Deficient HFD, and l-Cysteine and Vitamin D Co-Supplementation on Gene Expression of Musculoskeletal Markers in Mouse Skeletal Muscle

The skeletal muscle of HFD-fed mice showed attenuated myogenic markers (MyoD, Mef2c, and Csrp3) (Figure 1a), but there were no significant alterations in muscle dystrophy markers such as Atrogin1, Murf1, and Myostatin (Figure 1b). Only osteoprotegerin was downregulated in the RANK/RANKL/OPG system (Figure 1c). VD-deficient HFD-fed mice’s skeletal muscle showed downregulation of myogenic markers similar to those seen in the HFD-fed mice (Figure 1a). However, muscle dystrophy markers increased significantly in the skeletal muscle of the HFD-VD- group compared to those in the HFD group (Figure 1b). Compared to skeletal muscle in HFD-fed mice, the mRNA level of RANK/RANKL increased significantly in the HFD-VD- group, but the level of OPG was significantly downregulated in HFD-VD- group (Figure 1c).

Groups supplemented with l-cysteine or vitamin D alone showed a partially significant beneficial effect on markers such as MyoD, Mef2c, and OPG (Figure 1a,c). However, supplementation with LC or VD alone, or co-supplementation, significantly suppressed muscle the dystrophy markers, RANK, and RANKL in mouse skeletal muscle compared to results in the HFD-VD- group (Figure 1b). LC and VD co-supplementation more significantly alleviated myogenic markers and OPG (Figure 1a,c) in mouse skeletal muscle compared to results in the HFD and HFD-VD- groups, including those supplemented with LC or VD alone. 

These findings indicated that co-supplementing LC with VD enhanced the beneficial effects against musculoskeletal disorder marker gene expression in skeletal muscle compared to monotherapy supplementation with LC or VD.

### 3.2. Impact of High Glucose, Palmitate, and Inflammatory Cytokines on Musculoskeletal Markers 

Myotubes were exposed to high glucose-mediated glucotoxicity, palmitate-mediated lipotoxicity, and inflammatory cytokines to mimic low-grade inflammation models, which is observed in both obesity and diabetes. Glucolipotoxicity significantly downregulated the mRNA levels of the myogenic markers (MyoD, Mef2c, and Csrp3), and OPG, but the levels of dystrophy markers (Atrogin1, Murf1, and Myostatin), RANK, and RANKL were elevated (Figure 2a) compared to the control group. Inflammatory cytokines did not alter the level of myogenic markers. Proinflammatory cytokines such as MCP-1 and TNF elevated the expression of dystrophy markers, RANK, and RANKL (Figure 2b) compared to that in the control group. Collectively, this result indicates that glucolipotoxicity negatively affects the myogenic markers, dystrophy markers, and RANK/RANKL/OPG system, while inflammatory cytokines in vitro induce dystrophy markers, RANK, and RANKL. They may also contribute to musculoskeletal disorders. 

### 3.3. The Deficiency of Transsulfuration Pathway Key Genes GCLC and CSE (Knockdown) in Myotubes Affects Musculoskeletal Markers

The expression of myogenic markers and OPG was attenuated in the GCLC, and CSE siRNA treated myotubes (Figure 3a,b), but the levels of dystrophy markers, RANK, and RANKL increased significantly compared to those of the control group (Figure 3a,b). Altogether, these data demonstrate that inhibited flow in the rate-limiting sulfur-containing amino acid (l-cysteine) pathway leads to a deficiency in the physiological antioxidant glutathione (GSH) or hydrogen sulfide (H_2_S), which alone or synergistically, may alter the expression of the musculoskeletal marker genes. 

### 3.4. GSH and H_2_S Inhibit Muscle Dystrophy Markers and Positively Induce Myogenic Markers Genes

The possible beneficial effect of H_2_S or GSH on the expression of genes involved in myogenesis, muscle dystrophy, and the RANK/RAKL/OPG system was explored with the antioxidant precursors l-cysteine (a GSH/H_2_S precursor) or NaHS (an H_2_S donor) following the methods used in previous publications [4,6]. Results showed that compared to levels in the control group, the mRNA levels of myogenic genes and OPG significantly increased following LC or NaHS treatment, which also decreased dystrophy markers, RANK, and RANKL (Figure 4a). These responses to treatment with LC and NaHS indicate that GSH and H_2_S may directly or indirectly affect these genes and suggest that H_2_S and GSH may have a beneficial effect on muscle physiology. 

## 4. Discussion

Vitamin D (VD) is a nutrient essential for maintaining good bone health and improving muscle strength [1,2,3,37,38]. VD deficiency or insufficiency is associated with various musculoskeletal disorders [39,40,41]. After multivitamins, vitamin D, by itself, is the second-highest vitamin supplement consumed by the public for better health and delay or prevent musculoskeletal disorders [41,42,43,44]. However, controlled clinical studies show that VD alone supplementations’ have limited therapeutical benefits, despite the clinical association between VD deficiency and disease outcome [4,7,26,45]. This study examined the hypothesis that the co-supplementation of l-cysteine (LC) with VD is better compared to monotherapies with LC or VD at alleviating dyshomeostasis in the skeletal muscle of VD-deficient high-fat diet-fed (HFD-VD-) mice.

This study reports that LC+VD co-supplementation showed significant beneficial effects on vital myogenic markers such as MyoD, Mef2c, and Csrp3 in an animal model of HFD-VD-. Further, in vitro studies carried out in mouse myotubes demonstrated that, while H_2_S/GSH deficiency (oxidative stress or antioxidant deficient condition), high glucose, and palmitate (metabolic insults) decreased myogenic markers, inflammatory cytokines, such as TNF and MCP-1, did not affect the markers of myogenesis. Previously it has been shown that low physiological concentrations (a deficient state) of 1,25(OH)_2_D_3_ (active VD) induces transdifferentiation of muscle cells into adipose cells (adipogenesis), whereas higher (physiological and supraphysiological) concentrations attenuate this effect and promote myogenic cell differentiation [46,47]. Further, VD ameliorates fat accumulation with AMPK/SIRT1 pathway activation in myotubes [48,49]. Moreover, GSH depletion and chronic inflammation impair myogenic differentiation through redox-dependent and independent pathways, while these effects are reversible following NAC or GSH replenishment [50]. GSH and H_2_S levels affect the intracellular redox state. In vitro supplementation of LC and NaHS shows a positive effect on myogenic markers. Additionally, preclinical studies have shown that co-supplementation of LC + VD significantly reduced oxidative stress by boosting GSH and positively upregulating the VD-regulatory genes (VDBP/VD-25-hydroxylase/VDR) epigenetically in the liver of mice in a 25(OH)D deficiency mouse model [4,5,6]. GSH optimization, along with VD is a better approach to alleviate myogenic genes. Therefore, reduction or prevention of oxidative imbalance and VD in muscle is of vital importance for the maintenance of the myogenic pathway.

GSH is a critical redox factor that mitigates oxidative stress and oxidative damage to endogenous proteins, impairs cellular physiology and leads to the manifestation of the disease. Supplementation with the GSH rate-limiting amino acid precursor l-cysteine has been used successfully to improve the GSH status, VD metabolism genes, and lower the incidence of immune-metabolic syndrome [4,5,6,7,8]. In the present study, supplementation with LC, along with VD, suppressed skeletal muscle dystrophy markers such as Atrogin1, Murf1, and Myostatin and RANK and its ligand (RANKL). VD deficiency is associated with muscle atrophy. In vitro metabolic insults, exposure to inflammatory cytokines, and antioxidant deficient conditions (H_2_S/GSH deficiency) induced muscle dystrophy markers along with RANK/RANKL.

Conversely, OPG showed an inverse relationship, unlike RANK/RANKL. Previous studies have shown that the muscle content of Atrogin1 was the highest in patients deficient in VD and lowest in patients sufficient in it, whereas VD supplementation seemed to repel atrophic changes and systemic inflammatory markers [51]. Antioxidants maintain muscle homeostasis, whereas disturbed redox status, known as the major contributing factor towards atrophy. N-Acetyl cysteine (NAC) treatment reduces muscle atrophy through beneficial antioxidant, anti-inflammatory, and anti-fibrotic effects [52]. Further, GSH/NAC inhibits RANK-L induced osteoclastogenesis both in vitro and in vivo [53]. VD also maintains a regular rate of bone resorption and formation through the receptor activator of the nuclear factor κB (RANK)/RANK ligand (RANKL)/osteoprotegerin axis [54]. Our study shows that antioxidant supplementation with LC and NaHS suppresses muscle dystrophy markers, RANK/RANKL, and boosts the levels of OPG. Hence, LC + VD co-supplementation treatment was extremely effective against the myopathic changes contributing to pathophysiology.

## 5. Conclusions

In VD insufficiency/deficiency, muscle function and physical function may be impaired before clinical or biochemical signs of musculoskeletal disease are evident. Low circulating levels of both glutathione (GSH) and 25(OH)D are positively associated with metabolic syndrome and poor health in human subjects [4]. A possible explanation for the limited success of clinical trials with VD-alone could be the need to simultaneously optimize deficiencies in essential antioxidant nutrients such as LC along with VD. Therefore, an efficient novel therapeutic strategy would be using the combined nutraceuticals LC + VD, which could simultaneously antagonize cellular oxidative stress and inflammation and thus provide a better option for musculoskeletal system disorders than supraphysiological monotherapy with VD [7]. Current data obtained from preclinical studies in a 25(OH)D deficient mouse model suggest that LC + VD more effectively boosted the actions/efficacy of VD on musculoskeletal markers than monotherapies with LC or VD (Figure 4b). Our data support the need for a clinical trial of co-supplementation of LC with VD to achieve better health outcomes, including skeletal muscle functions. 

## Figures and Tables

**Figure 1 nutrients-12-03406-f001:**
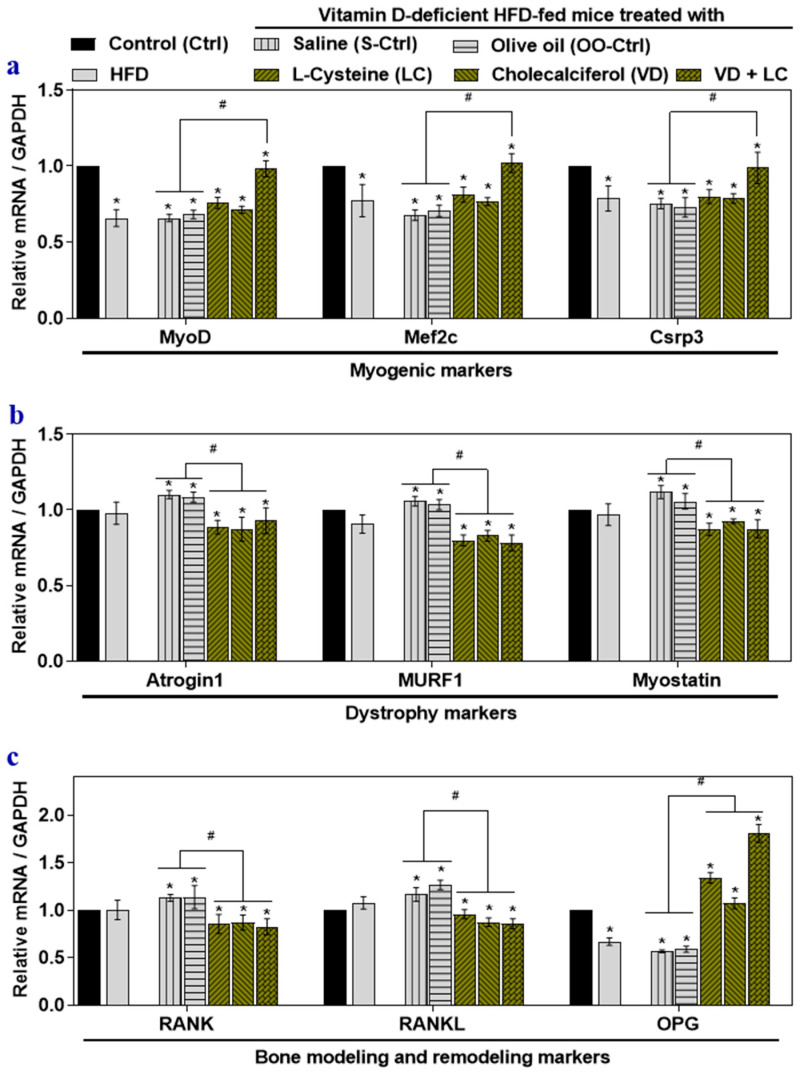
The effects of high-fat diet (HFD), vitamin D (VD)-deficient HFD (HFD-VD-), and l-cysteine and vitamin D co-supplementation on gene expression musculoskeletal markers in mouse skeletal muscle. Male C57BL/6J mice (5 weeks old) were fed with standard chow diet (Control; Ctrl), a high-fat diet (HFD), or a VD-deficient HFD for 16 weeks. Mice were gavaged with saline (S-Ctrl), Olive oil (OO-Ctrl), l-Cysteine (LC), Cholecalciferol (VD), or VD + LC during the last 8 weeks. The mRNA levels of myogenic marker genes: myoblast determination protein 1, myocyte enhancer factor 2C, and cysteine and glycine-rich protein 3 (MyoD, Mef2c, and Csrp3) (**a**); dystrophy marker genes: skeletal muscle-specific F-box protein, muscle RING-finger protein-1, and Myostatin (Atrogin1, Murf1, and Myostatin) (**b**); bone modeling and remodeling genes: receptor activator of nuclear factor-kB, receptor activator of nuclear factor-kB ligand, and osteoprotegerin (RANK, RANKL, and OPG) (**c**) were analyzed using qRT-PCR. Results are mean ± SEM (*n* = 4). Two-way ANOVA, followed by Dunnett’s multiple comparisons test, was performed between groups. Significance at *p* < 0.05: Asterisk symbol (*) represents a comparison between control (Ctrl) with all other groups, whereas the hash symbol (^#^) represents a comparison between HFD-VD- saline and olive oil control (S-Ctrl and OO-Ctrl) with LC, VD, VD + LC co-supplementation groups. MyoD: Myoblast determination protein 1, Mef2c: Myocyte enhancer factor 2C, Csrp3: Cysteine and glycine-rich protein 3, Atrogin1: skeletal muscle-specific F-box protein, Murf1: Muscle RING-finger protein-1, RANK: Receptor activator of nuclear factor-kB, RANKL: Receptor activator of nuclear factor-kB ligand, OPG: Osteoprotegerin.

**Figure 2 nutrients-12-03406-f002:**
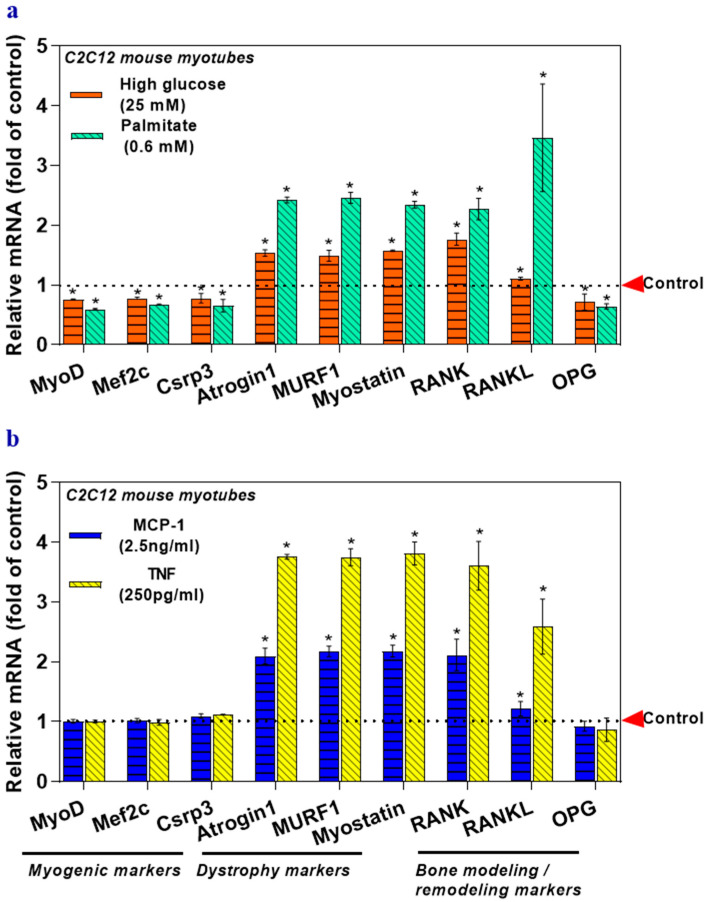
Glucolipotoxicity and inflammatory cytokines (MCP-1 and TNF) affect musculoskeletal markers in myotubes. Myotubes were treated with high glucose (25 mM) or palmitate (0.6 mM) for 24 h. Mannitol was used as an osmolality control (**a**). In another set of experiments, myotubes were treated with MCP-1 (2.5 ng/mL) or TNF (250 pg/mL) for 6 h (**b**). The mRNA levels of target genes responsible for myogenesis, muscle dystrophy, bone modeling, and remodeling (MyoD, Mef2c, Csrp3, Atrogin1, Murf1, Myostatin, RANK, RANKL, and OPG) were analyzed using qRT-PCR (**a**,**b**). Results are mean ± SEM (*n* = 3). Two-way ANOVA, followed by Dunnett’s multiple comparisons test, was performed between groups. A *p*-value of <0.05 for a statistical test was considered significant and represented as an asterisk symbol (*) compared with the control group. MCP-1: Monocyte Chemoattractant Protein 1, TNF: Tumor Necrosis Factor.

**Figure 3 nutrients-12-03406-f003:**
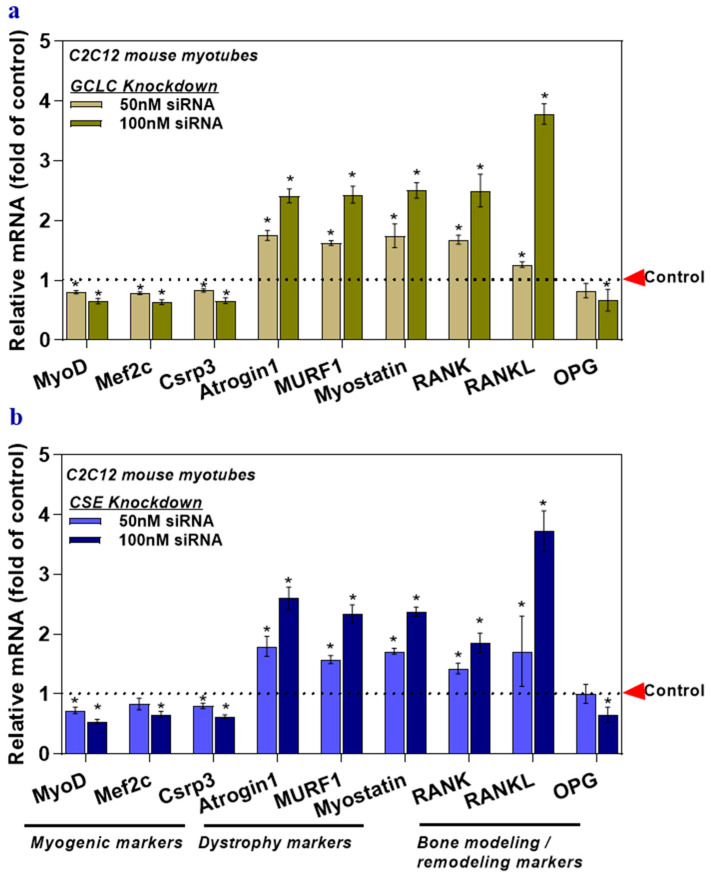
The deficiency of transsulfuration pathway key genes (GCLC and CSE knockdown) in myotubes affects musculoskeletal markers. Myotubes were transfected with GCLC siRNA (GSH deficient) (**a**) or CSE siRNA (H_2_S deficient) (**b**). Scrambled siRNA, served as a control. The mRNA levels of target genes responsible for myogenesis, muscle dystrophy, bone modeling, and remodeling (MyoD, Mef2c, Csrp3, Atrogin1, Murf1, Myostatin, RANK, RANKL, and OPG) were analyzed using qRT-PCR (**a**,**b**). Results are mean ± SEM (*n* = 3). Two-way ANOVA, followed by Dunnett’s multiple comparisons test, was performed between groups. A *p*-value of <0.05 for a statistical test was considered significant and represented as an asterisk symbol (*) compared with the control group.

**Figure 4 nutrients-12-03406-f004:**
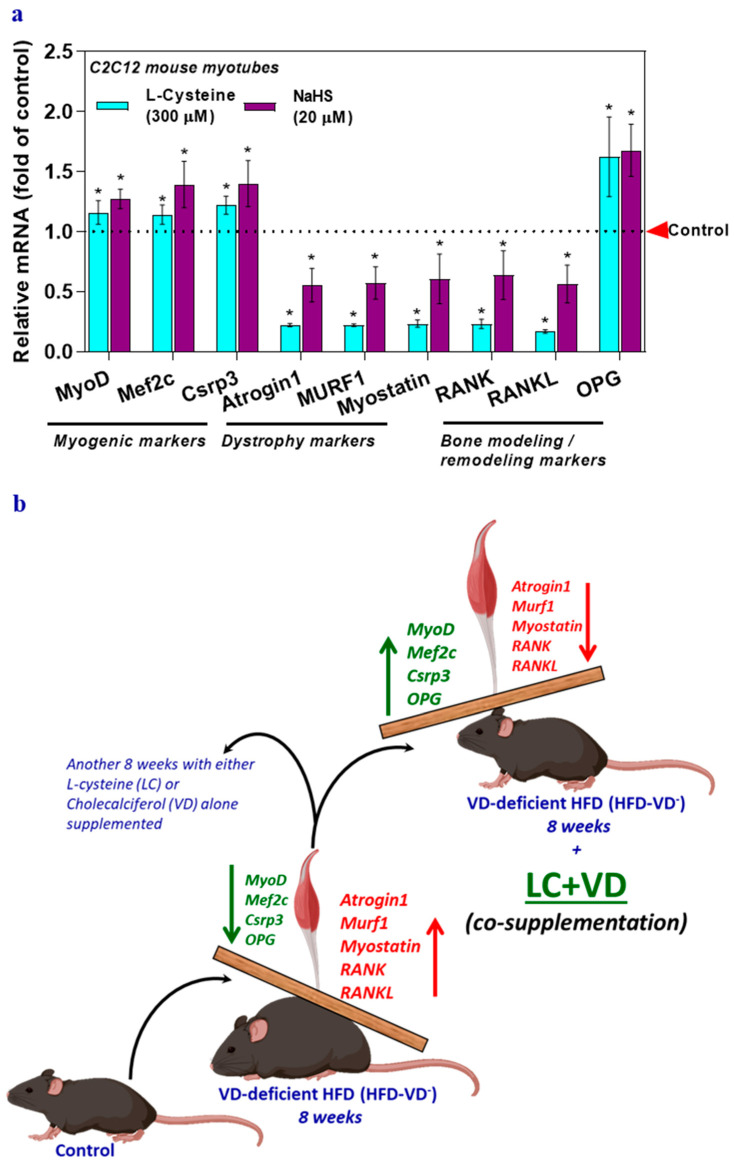
Glutathione (l-cysteine) or hydrogen sulfide (sodium hydrosulfide) supplementation alters musculoskeletal markers in myotubes. Myotubes were treated with either l-cysteine (LC; 300 µM) or sodium hydrosulfide (NaHS; 20 µM) for 6 h. The mRNA levels of target genes responsible for myogenesis, muscle dystrophy, bone modeling, and remodeling (MyoD, Mef2c, Csrp3, Atrogin1, Murf1, Myostatin, RANK, RANKL, and OPG) were analyzed using qRT-PCR (**a**). Results are mean ± SEM (*n* = 3). Two-way ANOVA, followed by Dunnett’s multiple comparisons test, was performed between groups. A *p*-value of <0.05 for a statistical test was considered significant and represented as an asterisk symbol (*) compared with the control group. Control animals were fed a healthy diet (Ctrl; lower in fat), while animals in the high-fat diet group were fed a high-fat diet (HFD) for a total of 16 weeks (not shown in scheme). The mice were maintained on a VD-deficient HFD (HFD-VD-) for 16 weeks (to mimic the VD-deficient condition). After the first 8 weeks, the mice were supplemented by oral gavage for another 8 weeks with either 5 mg LC/kg BW daily (LC), 67 IU VD/kg BW (VD) alone, or the same doses of LC + VD co-supplemented to HFD-VD-mice. The markers of Myogenic: MyoD, Mef2c, and Csrp3; Muscle dystrophy: Atrogin1, Murf1, and Myostatin; Bone modeling and remodeling: RANK, RANKL, and OPG were analyzed. Myogenic markers and OPG decreased in HFD-VD-mice muscle, whereas muscle dystrophy markers increased significantly. LC + VD co-supplementation to HFD-VD-mice ameliorate partially or entirely all the markers mentioned above at par with control groups (**b**).

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
