# Peer review of "l-Cysteine and Vitamin D Co-Supplementation Alleviates Markers of Musculoskeletal Disorders in Vitamin D-Deficient High-Fat Diet-Fed Mice"

_nutrients, 2020, doi:10.3390/nu12113406_

Round 1

Reviewer 1 Report

In this preclinical study the authors address the problem of vitamin D deficiency and skeletal striated muscle health. The introduction places the non-expert reader in a position to understand the technicalities of the work. The methodology is adequate to address the objective of the study. The methodology is clearly exposed from the housing of the animals to the design of the groups and the different techniques used. The results are clearly displayed both in the text and in the figures (Figure 4.B is especially useful). The figures are very well analyzed in the text.

Overall, it is a good job, well done and well presented. To improve it, I suggest some changes:

1.- Sometimes the authors take for granted knowledge that a non-expert reader does not know. A non-expert reader may find it helpful to know the bibliographic source of:

Line 40: “Skeletal muscle is the largest tissue in the body………….”

Line 40-51: the bases of the markers to be used in the study are given. However, no bibliographic reference is used. A chapter or book review can be helpful to the non-expert reader.

Line 249: “…. Vitamin D, by itself, is the second-highest vitamin supplement consumed by the public for better health and delay or prevent musculoskeletal disorders.”

Line 251: “controlled clinical studies show limited therapeutically benefits of VD alone supplementations despite the clinical association between VD deficiency and disease outcome.”

2.- The following sentence needs a bibliographic reference or must be eliminated:

Lines 97-99: This model (HFD and HFD-VD-) of dietary-induced insulin resistance created fasting hyperglycemia and hyperinsulinemia, glutathione, and VD deficiency and thus represented a reasonable model of the human condition.

3.- The first sentence of the results section is very confusing: HFD-fed mice (consumption of an HFD for 16 weeks) exhibited a metabolic phenotype similar to that of obese type 2 diabetic human subjects (Line 144) I do not understand how there can be a mouse metabolic phenotype be similar to that of humans. Perhaps you should define where there are specific similarities.

Reviewer 2 Report

This paper addressed the beneficial effect of co-supplementation of LC and VD on markers of musculoskeletal disorders in the skeletal muscle of VD-deficient high-fat diet-fed mice suggesting that LC+VD can be considered "an efficient novel therapeutic strategy which could simultaneously antagonize cellular oxidative stress and inflammation" and thus "can provide a better option for musculoskeletal system disorders than supraphysiological monotherapy with VD". In my opinion, the paper is well written and easy to follow. The hypothesis is clearly stated and the rationale for the hypothesis is well defined.

Below I express some questions

  1. The work is only performed in male mice. The authors should justify why they did not perform the experiments in both sexes and why the male sex was selected. 
  2. The authors indicate that mice were "housed for 1 week of acclimation in a temperature-controlled room (22±2 °C)". Thermoneutrality for mice is 30°C, this should be mentioned and the choice of 22±2 °C housing temperature discussed/justified. This is an important point because low temperatures can influence the expression of genes that code for proteins involved in muscle atrophy
  3. The images are not of high enough resolution
  4. The selection of the gastrocnemius for studying is still not clear. Why didn't you decide to analyze the effects of vc + lc co-supplementation on the soleus and EDL in order to evaluate both an example of a muscle with a higher content of slow fibers and a muscle with a higher content of fast fibers?
  5. In conclusion, you choose MyoD, Mef2c, Csrp3 as myogenic biomarkers. These are mostly markers of early differentiation state.  Why didn't you decide to analyze also late differentiation markers (such as Myogenin, myosins, Mrf4, dystrophin)? In this way you can see if muscle development is involved in all stages of differentiation and if lc + vd co-supplementation really has a beneficial role on muscle development. The authors have to confirm muscle differentiation involvement analyzing all differentiation states.

Round 2

Reviewer 2 Report

The authors satisfied my comments